# Sonidegib in Locally Advanced Basal Cell Carcinoma: A Monocentric Retrospective Experience and a Review of Published Real-Life Data

**DOI:** 10.3390/cancers15143621

**Published:** 2023-07-14

**Authors:** Gianluca Nazzaro, Valentina Benzecry, Maria A. Mattioli, Nerina Denaro, Giada A. Beltramini, Angelo V. Marzano, Emanuela Passoni

**Affiliations:** 1Dermatology Unit, Fondazione IRCCS Ca’ Granda Ospedale Maggiore Policlinico, 20122 Milan, Italy; 2Department of Physiopathology and Transplantation, University of Milan, 20122 Milan, Italy; 3Oncology Unit, Fondazione IRCCS Ca’ Granda Ospedale Maggiore Policlinico, 20122 Milan, Italy; 4Maxillofacial Surgery and Odontostomatology Unit, Fondazione IRCCS Ca’ Granda Ospedale Maggiore Policlinico, 20122 Milan, Italy; 5Department of Biomedical, Surgical and Dental Sciences, University of Milan, 20122 Milan, Italy

**Keywords:** basal cell carcinoma, locally advanced basal cell carcinoma, Hedgehog inhibitor, sonidegib, real-life

## Abstract

**Simple Summary:**

Basal cell carcinoma (BCC) is one of the most common malignancies worldwide. Some patients may develop locally advanced BCC with significant morbidity and with reduction in life quality. The employment of a Hedgehog inhibitor known as Vismodegib has already proven itself helpful in the management of laBCCs. Sonidegib is the most recently available drug approved for treatment of laBCCs that acts by inhibiting the Hedgehog pathway. Real-life data seem to show that efficacy and safety are similar to those already demonstrated in trials. Herein we report our experience with retrospectively collected data from laBCC patients treated with Sonidegib.

**Abstract:**

Basal cell carcinoma (BCC) represents the most common skin cancer and locally advanced BCC (laBCC) refers to an aggressive, large, infiltrative BCC that cannot be treated by surgery or radiotherapy. Sonidegib is a Hedghehog inhibitor (HHi) indicated for laBCC. This is a monocentric retrospective real-life study of laBCCs receiving Sonidegib treatment. Although Sonidegib is widely used, since its approval by Food and Drug Administration in 2015, only a limited number of real-life experiences have been reported. Eleven patients, including four patients diagnosed with Basal Cell Naevus syndrome, received treatment with Sonidegib for laBCCs. Seven (63.6%) patients experienced adverse events (AEs) but only three had to discontinue treatment and were therefore excluded from the following results. Four patients (50%) achieved complete clinical remission (CR); in all cases the remission was confirmed by biopsy. Partial response (PR) was found in three patients out of eight (37.5%). One patient out of eight (12.5%) showed a steady disease (SD). None of the patients showed signs of progression during treatment with HHi. Sonidegib showed the same efficacy in treating laBCCs as already seen in trials. All four patients suffering from Basal Cell Naevus syndrome achieved disease control by being treated with Sonidegib. Consequently, we strongly advise the joint management of laBCCs through a multidisciplinary team whenever feasible.

## 1. Introduction

Basal cell carcinomas (BCCs) represent the most common form of cancer globally, accounting for about 75–80% of keratinocyte carcinomas and showing a constantly increasing incidence [1]. First-line treatment for sporadic BCCs is surgery, whenever feasible. When surgery is contraindicated, based on a decision shared in a multidisciplinary team (MDT), radiotherapy and topical drugs, including imiquimod and 5-fluorouracil, and photodynamic therapy should be considered [2]. 

There is no accepted definition for locally advanced BCCs (laBCCs). Commonly, this term is used to refer to aggressive, large, deep tissue infiltrative, or recurrent BCCs [3]. LaBCCs include those BCCs that, either for patient- or disease-driven factors, are not considered suitable for curative treatment by surgery or radiotherapy [1]. 

Sporadic BCCs originate from a Sonic Hedgehog (HH) pathway dysfunction, including Patched 1 (PTCH1) gene mutations and following Smoothened (SMO)-driven activation of glioma-associated oncogene homologs (GLI) [3]. Physiologically, the HH pathway is very active during embryonic development; in the following phases PTCH1 constitutively exert an inhibitory action on SMO, inhibition that is lifted when an HH ligand binds to PTCH1. SMO, once effective, can activate GLI through an intracellular signaling cascade. A non-canonical Hedgehog signaling pathway might activate GLI transcription factors beyond SMO as well [4]. The mechanism is similar to that occurring in patients diagnosed with Basal Cell Naevus syndrome, an autosomal dominant genodermatosis presenting with early onset of multiple BCCs and skeletal, ophthalmologic, and neurologic abnormalities. GLI activation, usually determined by the PTCH1 mutation, is the underlying cause of this syndrome [5]. 

Hedgehog inhibitors (HHis) currently approved for the treatment of advanced BCCs are Vismodegib (150 mg daily) and Sonidegib (200 mg daily). Since they both prevent HH pathway activation by antagonizing SMO, they also share similar safety profiles [3]. Both these drugs showed efficacy in laBCCs; however, several side effects are commonly experienced, including muscle spasms, dysgeusia, and alopecia [6]. Recently, the HHi known as Patidegib was studied in the gel formulation at 4% and 2% concentration and completed the phase three trial showing promising results. This drug mechanism is the same as Vismodegib and Sonidegib, but its topical use may be very suitable in cases involving the elderly, and in those patients reporting unbearable side effects [7]. Although Sonidegib has been widely used since its approval by the Food and Drug Administration in 2015, only a limited number of real-life experiences have been reported. The data available in literature are summarized in Table 1 [8,9,10,11,12,13,14,15,16,17,18,19,20,21,22,23,24,25,26,27,28,29]. In those cases of laBCCs not responding to first-line systemic therapy with HHis, or when the treatment leads to the onset of unacceptable side effects (AEs), anti-programmed death 1 (anti-PD1) immune therapy with Cemiplimab may be considered [30,31].

In this observational retrospective article, we report real life experience with Sonidegib for advanced BCCs of a tertiary dermato-oncological department.

## 2. Materials and Methods

For the review of the literature, we used ClinicalTrials.gov to select the currently ongoing trials. In addition, we performed broad search in PubMed using key words “Sonidegib”, “BCC”, “basal cell carcinoma”, and “laBCC”, starting from the end of 2014. We excluded reviews and clinical trials. We excluded reports of real-life setting treatment with HHis when the results were not specifically attributed to Sonidegib and those cases in which Sonidegib was administered after Vismodegib failure or as part of a combination therapy. We identified 22 real-life reports addressing treatment outcome of laBCC and/or multiple BCC patients treated with Sonidegib. The review was updated to June 19th 2023.

Data from patients that underwent treatment with Sonidegib for advanced BCCs at our dermato-oncological department were retrospectively reviewed. Clinical responses were assessed according to what was previously reported in the literature [19,20]. A complete response (CR) corresponds to the clinical disappearance of the cancerous lesion; partial response (PR) was defined as a reduction in the tumor volume > 50%. Stable disease (SD) was used when the tumor volume is decreased in volume by ≤50% or increased by ≤20%. Progressive disease (PD) stands for an increase in tumor volume > 20%. Since all the patients included in the analysis presented with at least one laBCC, the assessment of the clinical outcome was performed on those. In the case of patients affected by Basal Cell Naevus Syndrome, we refer to the outcome of the laBCC which led to the indication to treat with Sonidegib. 

Histopathologic complete response was defined as the absence of tumor by punch biopsy. 

A multidisciplinary assessment was provided at the beginning of treatment and at disease clinical and instrumental evaluation. Our MDT consists of a core of specialists including dermatologists, ENT surgeons, maxillofacial surgeons, radiation oncologists, medical oncologists, and radiologists; on demand, we consult nutritionists, histologists, general surgeons, and plastic surgeons.

Blood count, renal and hepatic function, and creatine phosphokinase serum levels were assessed before starting therapy and monthly during treatment. 

The study was conducted in accordance with the ethical standards of the responsible committee on human experimentation (institutional and national), with the Helsinki Declaration of 1975, as revised in 2000, and with the Taipei Declaration. All patients provided written informed consent for publication of the material. Because of the retrospective nature of the study, only a notification to the Ethics Committee was requested.

## 3. Results

The case series consists of 11 patients who received treatment with Sonidegib for the presence of at least one laBCC. Six patients (54.5%) were males. The median age at the time of the introduction of Sonidegib was 54 years old (IQ range 32.75). Epidemiologic data and tumor-related information are summarized in Table 2. Three out of 11 patients (27.3%) discontinued treatment due to AEs. All of them suffered from muscle spasms. Two of them presented with both dysgeusia and fatigue, whereas weight loss was reported in one case only. Other AEs included alopecia barbae, nausea, and creatine phosphokinase increase. Their outcomes were excluded from the results. Patients’ treatment information, outcomes, and related AEs are summarized in Table 3. 

Four patients tested positive for PTCH1 gene mutation and were thus diagnosed with Basal Cell Naevus Syndrome. In all patients, Sonidegib was prescribed due to the presence of at least one laBCC. BCCs variants were distributed as follows: 1/8 micronodular type, 2/8 nodular types, and 3/8 infiltrative types. In two cases no data were available. Only one out of three infiltrative BCCs achieved CR. The remaining two both showed a partial response. The totality of the micronodular and nodular variants completely regressed both clinically and histopathologically. 

All the patients were given, whenever feasible, first-line surgical treatment. After surgery failure or recurrence of the tumor, they were started on Hedgehog inhibitors. Mean therapy duration was 11 months. 

Four patients (50%) achieved complete clinical remission (CR) (Figure 1); in all four cases the remission was confirmed by biopsy (Figure 2). Partial response (PR) was found in three patients out of eight (37.5%). One patient out of eight (12.5%) showed steady disease (SD). None of the patients showed signs of progression during treatment with HHi. In one case Sonidegib was administered as a neoadjuvant treatment before surgery. Another patient who achieved CR was treated with combined Sonidegib and radiation therapy. 

One out of four patients in our series who tested positive for PTCH1 mutation achieved CR. One of them, who only underwent treatment for 4 months, is currently showing no sign of progression (SD), whereas the remaining two patients achieved PR. Nine (81.8%) out of 11 patients experienced adverse events (AEs), and in six of them (66.7%) it was necessary to make dosage adjustments at some point. Mean therapy duration until AE onset was 3.2 months. The most common AEs were muscle spasms (5/8), dysgeusia (4/8), and hair unit disorders (6/8). The other reported AEs were fatigue (2/8), increase in CPK serum levels (1/8), weight loss (2/8), gastrointestinal disorders (2/8), and increase in serum lipase levels (1/11). Four patients out of 11 underwent treatment with Vismodegib before switching to Sonidegib. Vismodegib was suspended either for intolerance (3/4) or for loss of efficacy and disease progression (1/4). Interestingly, all the patients maintained at least a steady disease following the switch. However, two of them developed side effects similar to Vismodegib. Because of this, they were initially managed by halving the dosage, as recommended for drug use, but ultimately the persistent AEs led to the decision to discontinue the treatment and introduce Cemiplimab 300 mg every 3 weeks.

## 4. Discussion

The results regarding efficacy seem to be consistent with the literature, even though the percentage of patients achieving SD in our case series was higher than what was reported from real-life experiences. According to the literature data that we reviewed in Table 1, we calculated that 56.3% (67/119) of the cases achieved CR, whereas PR occurred in 37.8% (45/119) of the cases. We found only six (5.0%) reported cases of SD and none of progressing disease (PD). 

Although the number of patients who achieved CR, PR, and PD in our experience are similar to those reported in literature; the discrepancy between the number of patients experiencing SD (12.5% vs. 5.0%) when compared to real-life reports may be attributed to a subjective, experience- and clinical-based assessment of response and to the small size of our case series, including four patients previously treated with Vismodegib. LaBCC CR rates in real-life experiences happen to be much higher than expected. Although the Modified Response Evaluation Criteria in Solid Tumors (mRECIST) [32] used are stringent, according to the phase II BOLT trial (NCT01327053), only 4.5% of patients suffering from laBCC treated with 200 mg Sonidegib achieved complete clinical remission and about 56.1% of patients with laBCC showed a response to treatment at all. The percentage of laBCC patients achieving CR rose to 21.1% when the BCC-RECIST-like criteria were applied. [33] According to a review of both clinical trials and real-life reports, the pooled percentage of patients who had at least stable disease is 94.9%. [34] This very high response expectation has been satisfied in this case series, with none of the patients showing disease progression during treatment. 

Villani et al. [20] analyzed the disease response to Sonidegib in 18 patients and observed no differences between the 24-week treatment outcome of the most and least aggressive BCC variants. However, the authors pointed out that, among infiltrative tumor subtypes, CR rate was accelerated to some extent, showing improvement by the end of the 12-week outcomes. Similar results were obtained by Fosko et al. while investigating whether a difference exists between different histotypes of BCC being treated with Vismodegib. [35] The results of our small case series show that, although both nodular BCCs completely regressed, only 1/3 infiltrative subtypes achieved CR. Interestingly, this very case showed a complete response within the first two months of therapy. Nonetheless, the group of patients enrolled in this retrospective analysis is too small to draw conclusions of statistical significance. 

Similar to what was reported by both trials and real-life experiences, the AEs commonly experienced by patients receiving Sonidegib in this real-life setting were muscle spasms, alopecia, and dysgeusia (Table 4). Nguyen et al. found that HHis-related AEs occurring in real life with higher frequency than reported in trials were weight loss, fatigue, and nausea. [34] In our case series, fatigue was reported by 25% of patients, and 2 out of 8 patients experienced significant weight loss. To date, none of the patients included in this case series have experienced the onset of squamous cell carcinoma following therapy with Sonidegib. It is noteworthy that, in one case the blood tests showed an increase in serum lipase levels that was managed by taking a drug holiday, and in another patient, a CPK increase required halving of the dosage. Reportedly, 25% of patients treated with Sonidegib discontinue the treatment because of the onset of unbearable AEs. Our study showed similar results, with 27.3% dropouts due to intolerance. During the phase II trial BOLT, the percentage of patients treated with Sonidegib that had discontinued treatment following the onset of unacceptable AEs was similar (30.4%) [33]. In addition, about half (66.7%) of the patients in this case series required a dosage adjustment at some point, either by halving the dose or by taking a drug holiday. Surprisingly, in the real-life literature reports summarized in Table 1, only 18/119 cases (15.13%) of discontinuation due to intolerance have been reported. 

Both intrinsic and acquired resistance have been reported in clinical studies. Two mechanisms have been identified: some mutations may modify the SMO binding site; whereas in other cases, the mutations affect downstream molecules, making the HH pathway constitutively active. In a group of patients with HHi resistance, it was not possible to identify SMO mutations [36]. According to the reported experience of nine patients with either acquired or intrinsic Vismodegib resistance, most of them did not show any response to Sonidegib. Only three out of a group of nine patients showing resistance maintained the SD status until surgery was performed on two of them [37]. Consistently, both patients in our series showing PR had been previously treated with Vismodegib.

According to a review [34], data from 257 patients diagnosed with Basal Cell Naevus Syndrome and treated with HHis are available across the literature: 83.3% of them achieved at least SD, while CR was observed in 45.9%. Only seven of them were taking Sonidegib. In our case series, four patients with a diagnosis of Basal Cell Naevus Syndrome were included and none of them showed any sign of disease progression. As summarized in Table 2 and Table 3, three of them had a good response to treatment with Sonidegib and one patient is currently in a state of clinical stability. 

Recently, Moreno-Arrones O.M. et al. published results from a Spanish national registry, including 82 laBCC patients treated with Sonidegib. They reported no difference between the results experienced by Basal Cell Naevus Syndrome patients (10/82) and the rest of the subjects. The Spanish group reported that 29.3% of patients achieved CR, whereas 52.4% showed PR. Interestingly, they reported that 6.1% of cases showed clinical progression of disease. These results reflect a somewhat minor efficacy of the treatment when compared to other real-life experiences. Probably, the inclusion of patients previously treated with Vismodegib (19.5%) is the explanation for the poorer trend of response to Sonidegib; of these patients, only 35.7% showed improvement with Sonidegib [38].

Since recurrence after Vismodegib was reportedly found in about 31% of patients who had exhibited CR [39] and considering the risk of developing HHis resistance [36], we strongly advise in favor of HHis employment as a neoadjuvant treatment before surgery, whenever feasible, to prevent relapse and to aim for curative treatment [40]. A phase II non-randomized pilot study is ongoing investigating Sonidegib in a neoadjuvant setting for laBCC, followed by surgery or imiquimod (NCT03534947).

Moreover, another possible treatment strategy, as in the case of one patient of this case series, may be the combination of radiotherapy and HHi [21,40]. Interestingly, especially for those in which a neoadjuvant approach is not possible and for patients who refuse surgical procedures, a study investigating the use of a Vismodegib maintenance dosage of 150 mg per week showed promising results [41]. Moreno-Arrones O.M. et al. found the onset of fewer AEs and a comparable efficacy when Sonidegib 200 mg was administered every other day [38]. Every one of the patients in this case series has been periodically assessed in a multidisciplinary setting formed by oncologists, maxillofacial surgeons, and dermatologists, among others, according to the most recent European consensus on BCC management guidelines [2]. The main limitations of this case series are the small number of patients and the mostly clinically based response to treatment evaluation. Moreover, the short treatment duration of some of the patients included in this case series must be considered. 

**Table 4 cancers-15-03621-t004:** Comparison between adverse events reported in literature as associated with treatment with Vismodegib and Sonidegib [33,34,36,37,39,40,41,42].

	Vismodegib	Sonidegib
Meal	No interference	On empty stomach
Drug interactions	+ (minor substrate of CYP2C9 and CYP3A4)	+ (avoiding the CYP3A inhibitors)
AE G ≤ 2	43%	54%
AE G ≥ 3	56%	43%
Muscle spasms G ≥ 3	6%	3%
Alopecia G ≤ 2	66%	50%
Diarrhea G ≥ 3	3%	1%
Weight loss G ≥ 3	9%	5%
Fatigue G ≥ 3	5%	1%
CK G ≤ 2	NR	24%
Dysgeusia G ≤ 2	56%	44%
Nausea G ≤ 2	33%	38%
ORR LABCC	60%	71%
ORR M BCC	49%	23%
DOR (median months) LABCC	26	16
DOR (median months) MBCC	15	18

Legend. ORR, overall response rate; LABCC, locally advanced basal cell carcinoma; MBCC, metastatic BCC; AE, adverse events; DORs, duration of response. Toxicities were reported in BOLT study and ERIVANCE study according to CTCAE (common toxicities criteria adverse events). NR, not reported; CK, creatin kinases.

## 5. Conclusions

The efficacy in treating laBCCs shown by Sonidegib is higher than that demonstrated in trials and similar to that reported in real-life settings. Real-life reports seem to experience a higher rate of CR than clinical trials. Conversely, dropouts due to intolerance happen to be more frequent in research settings when compared to real-life ones. Similar to what has been reported in trials, a large proportion of patients in this case series had to discontinue treatment due to the onset of unbearable AEs. Future efforts should be focused on finding successful strategies to achieve long-term results either by reducing the maintenance dosage or by adopting combined or neoadjuvant treatment approaches. The main limitations of this case series are represented by the small number of patients retrospectively included and their different treatment duration times. The latter factor could represent a confounding factor.

## Figures and Tables

**Figure 1 cancers-15-03621-f001:**
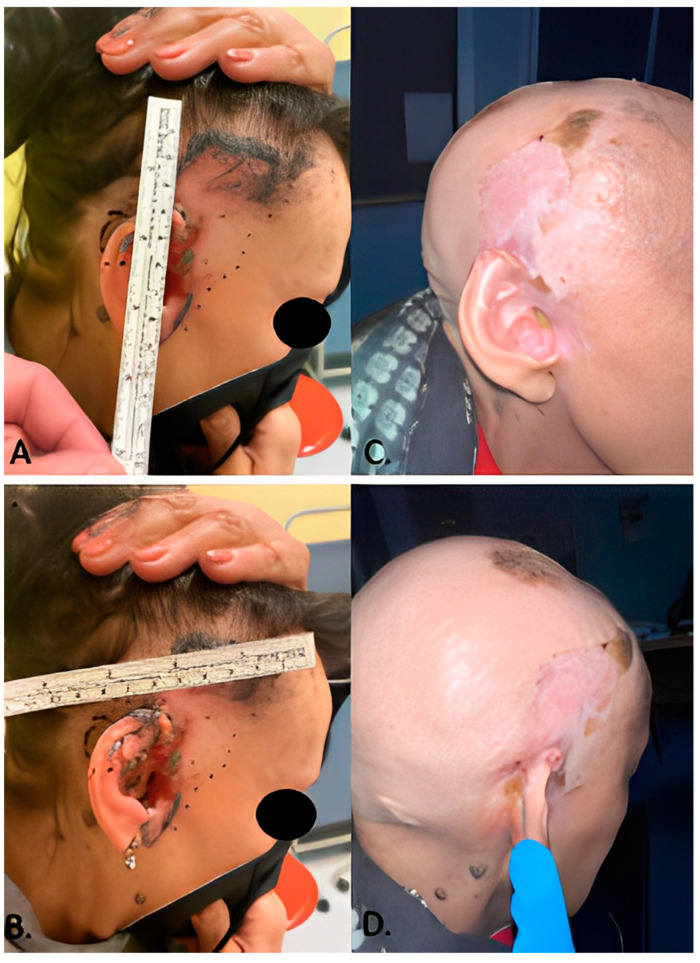
(**A**,**B**) patient number 2 photographed before starting treatment with Sonidegib 200 mg daily. (**C**,**D**) the same patient after 1 year of therapy. The complete remission of the locally advanced BCC was histologically confirmed, and Sonidegib was discontinued.

**Figure 2 cancers-15-03621-f002:**
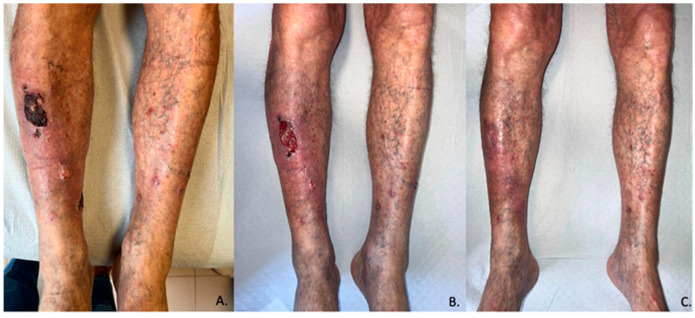
Pictures of patient number 11 of this case series. (**A**–**C**) respectively represents T0 (before treatment), T1 (1 month), and T3 (3 months).

**Table 1 cancers-15-03621-t001:** Real life experience with Sonidegib in advanced BCCs and/or multiple BCCs.

Year	Authors	N	Age	(n) Sex	Location	Previous Treatments	Sonidegib Duration	(n) Outcome	AEs (n) Dropouts
2020	Conforti C. et al. [8]	1	80	F	Multiple la BCCs	Surgery	3 m	CR	none
2020	Hou X. et al. [9]	1	73	M	Periocular laBCC	-	6 m	CR	MS, vomiting
2020	Villani A. et al. [10]	1	67	M	laBCC	Surgery	6 m	CR	None
2021	Conforti C. et al. [11]	1	90	-	Multiple + anal and rectal la BCC	-	>5 m	CR	MS, ↑ CPK
2021	Colnè J. et al. [12]	4	94 88 70 70	F F F F	Periocular laBCC Periocular laBCC Periocular laBCC Periocular laBCC	-	11 m 24 m 24 m 3 m	CR PR PR PR	Alopecia, dysgeusia, GI, MS, weight loss 1 dropout
2021	De Giorgi V. et al. [13]	2	-	-	Periocular La BCC	Surgery	-	(2) PR	Asthenia, CPK elevation, MS 0 dropouts
2021	Fania L. et al. [14]	1	64	M	laBCC	Surgery	8 m	CR	Alopecia
2021	Hoffmann V. et al. [15]	1	51	M	Multiple BCCa	Surgery	9 m	CR	Alopecia totalis, dysgeusia, MS, ↑ CPK, xerostomia, weight loss
2021	Moscarella E. et al. [16]	5	-	(4) M (1)F	laBCC laBCC	Surgery (2) RT + surgery	(1) 9 m (3) 6 m (1) lost	(1) CR (3) PR (1) lost	Alopecia, Fatigue, transaminase increase, MS 1 dropout
2021	Rokohl A. et al. [17]	1	-	-	Multiple + periocular BCC	Surgery	6 m	CR	-
2021	Tarantino V. et al. [18]	1	71	F	Multiple BCCs	-	6 m	CR	CPK increase, dysgeusia, MS 0 dropouts
2021	Toffoli L. et al. [19]	9	81 89 90 48 78 52 83 59 75	F M M M M F F F F	Multiple + la BCCs laBCC Multiple + laBCCs Multiple + laBCCs Multiple + la BCCs Multiple BCCs laBCCs laBCCs Multiple BCCs	Surgery ECT + PDT - Surgery - Surgery Surgery - -	21 m 9 m 5 m 7 m 7 m 25 m 5 m 4 m 5 m	PR PR CR PR PR PR PR PR PR	Alopecia, dysgeusia, GI, bone pain, muscle spasms 0 dropouts
2021	Villani A. et al. [20]	18		(16) M (2) F	laBCC laBCC	-	6 m	12CR 4PR 2SD	None
2021	Wang K. et al. [21]	1	85	M	laBCC	-	8 m	PR	-
2021	Weis J. et al. [22]	1	78	F	laBCC	Surgery Cemiplimab	8 m	CR	None
2022	Camela E. et al. [23]	1	77	F	laBCC	-	10 m	CR	MS, dysgeusia, alopecia
2022	Leow L. J. et al. [24]	10	54 65 62 67 77 46 37 49 56 59	F M M M F F F F M M	laBCC laBCC laBCC laBCC laBCC laBCC laBCC laBCC laBCC laBCC	-	12 m 6 m 3 m 7 m 3 m 8 m 2 m 4 m 3 m 9 m	CR CR CR CR PR PR CR CR CR CR	Dysgeusia, MS, hyperhidrosis, alopecia, nausea, loss of appetite, fatigue, arthralgia 0 dropouts
2022	Puig S. et al. [25]	2	79 69	F M	laBCC laBCC	Surgery RT -	8 m 6 m	CR CR	Ageusia, MS, weight loss, alopecia 0 dropouts
2022	Toffoli L. et al. [26]	2	59 83	F F	laBCC laBCC	- Surgery	4 m 6 m	PR CR	MS, nausea, loss of appetite, bone pain 0 dropouts
2022	Trabelsi S. et al. [27]	1	63	M	Multiple + periocular laBCC	-	2 m	CR	-
2022	Villani A. et al. [28]	54	77.8	-	Multiple + laBCCs	-	7.2 M	(29) CR (21) PR (4) SD	MS, dysgeusia, weight loss, alopecia 16 dropouts
2023	Piccerillo A. et al. [29]	1	69	F	laBCC	-	7 m	CR	MS, alopecia
Total	22	119	-	-	-	-	-	(67) CR (45) PR (6) SD (1) lost	18 dropouts

Legend. laBCC (locally advanced BCC); MS (muscle spasms); m (months); CR (complete remission); PR (partial remission); SD (steady disease); RT (radiant therapy); PDT (photodynamic therapy); ECT (electrocoagulation therapy); ↑ (increase).

**Table 2 cancers-15-03621-t002:** Epidemiological and clinical characteristics of patients treated with Sonidegib.

PT	Age	Sex	Comorbidities	Type of BCC	Localization	N D (Larger)	PTCH1 Mutation
1	53y	M	Cleft lip and palate Longitudinal sickle calcification	Multiple BCCs laBCC	Head and neck Trunk and limbs Scalp	>100 BCCs >10 BCCs 5 × 6 cm	+
2	74y	F	Dyslipidemia Arterial hypertension	La BCC	Temporal area, right ear	10 × 9 cm	Not tested
3	88y	M	Diverticulosis Umbilical hernia Cholecystectomy Abdominal aortic aneurysm Arterial hypertension Chronic renal failure Transient ischemic attack	laBCC	Left ear	-	Not tested
4	45y	F	-	Multiple BCCs laBCC	Head and neck, trunk Right ear	>10 BCCs >10BCCs -	+
5	45y	F	-	laBCC	Right cheek and periocular	5.3 cm	Not tested
6	70y	M	-	laBCCs Multiple BCCs	Scalp Head and neck	- >7BCCs	Not tested
7	71y	F	Diabetes mellitus 1 Meningioma	Multiple BCCs LaBCC	Head and neck Scalp	- 10 × 5 cm	Not tested
8	79y	M	Arterial hypertension Hypothyroidism Mild cognitive impairment	laBCC	Right shoulder, hemithorax, and hip	-	-
9	54y	M	Cleft lip and palate	laBCC Multiple BCCs	Inferior eyelid + nose Head and neck	-	+
10	30y	F	-	Multiple BCCs laBCCs	Head and neck Scalp	-	+

Legend. PT (patients); BCC (basal cell carcinoma); N (number); D (diameter); y (years); F (female); M (male).

**Table 3 cancers-15-03621-t003:** Patients’ treatment information, outcomes, and related AEs.

Pz	Previous TP	HHi	Following TP	CR	PR	SD	PD	TP Duration	AEs	Dosage Adjustments	AEs Onset	Path
1	Surgery	Sonidegib	Surgery	+ (biopsy)	-	-	-	14 m (ongoing)	Madarosi CPK ↑ Muscle spasms	200 mg/2 days 6 weeks holiday	10 m	-
2	RT 5FU RT	Sonidegib	-	+ (biopsy)	-	-	-	14 m (ongoing)	Weight loss Muscle spasms Alopecia Dysgeusia	-	4 m	Micronodular
3	Surgery	Sonidegib	-	+	-	-	-	6 m	Muscle spasms Dysgeusia fatigue Alopecia barbae	200 mg/2 days discontinuation	1 m	-
4	Surgery	Sonidegib	-	+ (biopsy)	-	-	-	20 m	Muscle spasms fatigue Dysgeusia Alopecia GI, Lipase ↑ Weight loss	200 mg/2 days 3 weeks holiday 200 mg/2 days	2 m	Nodular
5	Vismodegib	Sonidegib + RT	-	-	+	-	-	18 m (ongoing)	Ageusia, Telogen	-	-	Sclerosing and infiltrative
6	Surgery RT Vismodegib RT	Sonidegib	Imiquimod RT Cemiplimab	-	-	+	-	2 m	Muscle spasms Dysgeusia Nausea Fatigue Weight loss	200 mg/2 days discontinuation	1 m	Superficial and ulcerated
7	Surgery RT Vismodegib	Sonidegib	Cemiplimab	-	-	+	-	6 m	Cpk ↑ Muscle spasms	Drug holiday (3 w) 200 mg/2 days discontinuation	2 m	-
8	Surgery	Sonidegib	-	+ (biopsy)	-	-	-	10 m (ongoing)	Ageusia, telogen	-	4 m	Nodular
9	Imiquimod, Surgery Vismodegib	Sonidegib	-	-	+	-	-	4 m (ongoing)	-	-	-	Infiltrative
10	Surgery PDT	Sonidegib	-	-	-	+	-	4 m (ongoing)	Muscle spasms Fatigue	-	3 m	-
11	-	Sonidegib	-	+	-	-	-	4 m (ongoing)	Muscle spasms Alopecia	200 mg/2 days	2 m	Sclerosing and infiltrative

Legend. Tp (therapy); RT (radiotherapy); 5FU (5-Fluorouracil); m (months); CR (complete response); PR (partial response); SD (steady disease); PD (progression of disease); AEs (adverse events); ↑ (increase).

## Data Availability

The data presented in this study are available on request from the corresponding author.

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
