# Peer review of "Sonidegib in Locally Advanced Basal Cell Carcinoma: A Monocentric Retrospective Experience and a Review of Published Real-Life Data"

_cancers, 2023, doi:10.3390/cancers15143621_

Round 1
Reviewer 1 Report (Previous Reviewer 3)
The manuscript was well-revised. In the finalized PDF without tracked changes, there is a disconnect in the top row of Table 3 where the column identifiers should be. It is also missing Table 2, but this was available in the tracked changes document, so I was able to review it there.
Author Response
Dear reviewer,
We thank you for your kind suggestions. We tried to redact the manuscript at our best and to fix the tables problems.
Kind regards
Reviewer 2 Report (Previous Reviewer 1)
I want to thank the Authors to address the previous comments. In my opinion, now the manuscript is more understandable and the introduction provides sufficient background to the research.
Still some minimal concern, as:
- Page 1, line 40, abbreviation HHi not previously explained;
- Page 8, line 1, updatated should be edited to updated;
- Page 12, line 369, it seems that ref 10 and ref 11 are the same.
There are also some other typoes along the paper, so I suggest to the Authors to carefully read it again and edit all these minor mistakes.
The English language is acceptable.
Author Response
Dear reviewer,
we deeply thank you for your kind suggestions and revised the paper for typoes following your advice.
Kind Regards
This manuscript is a resubmission of an earlier submission. The following is a list of the peer review reports and author responses from that submission.
Round 1
Reviewer 1 Report
This interesting paper concerns an observational retrospective study reporting the real-life experience of a tertiary dermato-oncological Department in Italy in the treatment of locally advanced basal cell carcinomas with the hedgehog inhibitor sonidegib. The topic of the hedgehog pathway inhibitors is something of great actuality, which begun with the approval of vismodegib and is still developing with the new molecule patidegib. Although sonidegib is widely used since its approval, only a few real-life experiences have been reported in literature, and this is the strenght of this study.
The paper is well-written and the results are in accordance with those of the trials previously reported in literature. The following points should be addressed:
1) Improving the introduction, better explaining the hedgehog pathway inhibitors and reporting the existence of a newer molecule (patidegib), would make the paper more attractive.
- Cosio T, Di Prete M, Di Raimondo C, et al. Patidegib in Dermatology: A Current Review. Int J Mol Sci. 2021 Oct 3;22(19):10725. doi: 10.3390/ijms221910725. PMID: 34639065; PMCID: PMC8509734.
2) Table 1 is useful to summarise what has been already reported in literature. The last line is without any reference to previous study. Where do the results come from? It is probably the total of the lines above, but also in that case it should be specified.
3) In the Results section it is stated that "In one case Sonidegib was administered as a neoadjuvant treatment before surgery". The use of hedgehog pathway inhibitors as neoadjuvant therapy is something suggested by experts in this field, but not yet well documented in the literature. So, it would be interesting to report also the response of the disease to the treatment (with clinical pictures, if available) and if the subsequent surgery was less destructive than it could have been without neoadjuvant therapy.
Moderate editing of English language is required especially in the abstract and introduction sections.
Author Response
Dear reviewer, thank you for your kind suggestions. We tried to improve our work following your advice.
- We tried to better explained the hedgehog pathway and mentioned the newest HHis patidegib which is currently under investigation.
- We specified that the last line of Table 1 reported the the total of the lines above; we thank you for making us notice the missing data.
- Sonidegib was administered as neoadjuvant treatment in the case of patient number one, however surgery was performed at a different facility. We tried to recover pre and post surgery pictures without success. Regretely, we cannot report details of the sugery since it was not performed by our equipe. This is the reason behind the choice to keep this very interesting topic as a mariginal one in this manuscript. In our intention, since the patient is still under treatment and follow up at our facility, led by his exemple, we wanted to outline the use of HHis as a neoadjuvant treatment as a potentially very effective strategy which is currently under investigation, and which we are willing to adopt in selected cases and once the trials will confirm its advantages. Kind Regards.
Reviewer 2 Report
This is a report from experience with Sonidegib in locally advanced basal cell carcinoma (laBCC) and a review of previously published real-world data. Clinical verification of results shown in previous studies is important and such studies are welcomed. However, there are some points related to this study that need to be mentioned. A change of the title in the direction of "case studies" would perhaps better reflect the content of the article. As the report also states it provides a review of previously published data in the area, it would be desirable to have a description of the criteria for selection ( search strategies and databases) that were used to select the reference articles. The authors state that a total of 11 patients with laBCC were treated, but from Table 2 it appears that only 10 of the patients had laBCC. Since the use of Sonidegib is not an experimental treatment, the results presented would have been more significant if a larger material had been included. In addition, the observation time for more than 50% of the reported cases is ≤ 6 months. In the report, complete response (CR) is defined as clinical disappearance of the cancer lesion. However, results are presented for patients (not lesions) even though several patients appear to have had multiple lesions. Parts of the conclusion section contain information that seems better suited in the discussion section. Working in interdisciplinary teams is important, but this topic was not studied and should therefore not be included in the conclusion.
Author Response
Dear reviewer, thank you for your kind suggestions. We tried to improve our work following your advice.
- We changed the title to be more aligned with the contet of the manuscript, as suggested.
- We specified the details of the databases and strategies used to gather the available information from the literature and we broaded our research in order to offer a accurate overview of the current state of art.
- We specified that, in case of patients affected by Basal Cell Naevus Syndrome, the indication to treatment with Sonidegib was based on the presence of at least one laBCC and that the outcome was assessed on that same tumoral lesion. We specified the presence of one laBCC in patient number one, Table 1. This data was missing and we really thank you for kindly pointing it out so we could correct the mistake.
- We updated the results to the date of this revision, aiming to improve the utility of this case series to the literature. As expected the results aligned even more closely with those found in the literature. We do hope that our findings and the analysis of the literature we performed represent something useful for the health professionals who are approaching the treatment of laBCCs by the use of HHis.
- The results presented refer to the laBCC for which the patients started the treatment. Many of them also presented multiple BCCs but their eventual response to treatment was not taken into account in the results.
- We revised the conclusions and discussion section as requested.
- We eliminated from the conclusions the part inherent to the multidisciplinary team working.
Kind Regards
Reviewer 3 Report
page1 line 21 change to "approved for treatment of laBCCs"
line 24 change infiltration to infiltrative
line 25 change to "indicated for laBCC"
line 35 "achieved disease control"
line 36 rephrase and correct "we strongly advise the joint management of laBCCs through a ... "
line 134 change to radiation
Please review and revise the whole manuscript for grammatical corrections.
For Patient 1 with a PTCH1 mutation and >100 BCCs, how was CR identified? Please specify if this patient demonstrated complete clinical resolution of all identifiable BCCs or just of the larger tumor noted.
As noted on lines 225-231, two patients have started treatment <2 months ago. It would be worthwhile to update the response of these patients at a before publication.
This is a well-presented case series reporting real-life outcomes of sonidegib for the use of advanced BCCs. I do feel the limited number of patients and the short follow up in several patients (<3 months) limits the utility of this publication stand-alone. It could however still be a useful addition to the literature and included in future reviews on the subject matter.
Please review and revise the whole manuscript for grammatical corrections.
Author Response
Dear reviewer, thank you for your kind suggestions. We tried to improve our work following your advice.
- We made the correction as kindly requested
- We reviewed the manuscript for grammatical corrections
- We specified that, in case of patients affected by Basal Cell Naevus Syndrome, the indication to treatment with Sonidegib was based on the presence of at least one laBCC and that the outcome was assessed on that same tumoral lesion. We specified the presence of one laBCC in patient number one, Table 1. This data was missing and we really thank you for kindly pointing it out so we could correct the mistake.
- We updated the results to the date of this revision, aiming to improve the utility of this case series to the literature. As expected the results aligned even more closely with those found in the literature. We do hope that our findings and the analysis of the literature we performed represent something useful for the health professionals who are approaching the treatment of laBCCs by the use of HHis.
Kind Regards